# Assessing the validity of post-discharge readmission and mortality as a composite outcome among newborns in Uganda

Abhroneel Ghosh[1], Vuong Nguyen[2], J. Mark Ansermino[2,3], Yashodani Pillay[2,3], Angella Namala[4,5], Joseph Ngonzi[6], Nathan Kenya-Mugisha[7], Niranjan Kissoon[2,8], Matthew O. Wiens[2,3,7] *

1 Indian Statistical Institute, Kolkata, West Bengal, India, 2 Institute for Global Health, BC Children's and Women's Hospital, Vancouver, British Columbia, Canada, 3 Department of Anesthesiology, Pharmacology & Therapeutics, University of British Columbia, Vancouver, Canada, 4 Jinja Regional Referral Hospital, Jinja City, Uganda, 5 Ministry of Health for the Republic of Uganda, Kampala, Uganda, 6 Dept of Obstetrics and Gynaecology, Mbarara University of Science and Technology, Mbarara, Uganda, 7 Walimu, Kampala, Uganda, 8 Department of Pediatrics, University of British Columbia, Vancouver, Canada

* Matthew.Wiens@bcchr.ca

## Abstract

### Background

Composite outcomes, which include mortality and readmission rates, are often used in risk prediction models following hospital discharge when event rates for the primary outcome of interest, mortality, are low. However, greater readmission rates may result in reduced mortality making interpretation of the composite outcome difficult. We assess the usefulness of a composite outcome of post-discharge readmission and mortality as a target outcome in this context.

### Methods

This was a secondary analysis of data collected among mothers and their newborn(s) admitted for delivery at two regional referral hospitals in Uganda. Six-week post-discharge mortality (all-cause) and readmission in newborn infants were analyzed using a competing risk framework. The Sub distribution Hazard Ratios (SHRs) were compared across predictor variables to examine the relationship between the two outcomes.

### Results

A total of 6040 newborns with complete six-week follow-up were enrolled, of whom 50.6% were male and 64% of mothers delivered via caesarean section. Thirty-five (0.58%) infants died within the six-week follow-up period and 241 (3.99%) were readmitted. Of the 206 predictors, 81 had a consistent association with both outcomes. These include a higher weight (SHRs: 0.14, 0.68) and length of the baby

**Data availability statement:** Data cannot be shared publicly because of the sensitive nature of the clinical patient data and potential risk for re-identification of research participants. This is also required by the Makerere University School of Public Health Institutional Review Board. Data are available from the Pediatric Sepsis Data CoLaboratory (Sepsis CoLab) (contact via sepsiscolab@bcchr.ca) for researchers who meet the criteria for access to confidential data. The data underlying the results presented in the study are available from Borealis: https://doi.org/10.5683/SP3/QP5WYE.

**Funding:** The Mitacs Globalink Research Internship provided the opportunity for Abhroneel Ghosh to conduct research with the team at the Institute for Global Health.

**Competing interests:** The authors have declared that no competing interests exist.

(SHRs: 0.85, 0.91). However, 125 variables depicted an association in opposing directions which may be linked to social and financial barriers to care-seeking. These include a travel time to the hospital of greater than 1 hour (SHRs: 1.4, 0.28).

## Conclusion

While mortality is unequivocally a negative outcome, readmission may be a positive outcome, reflecting health seeking, or a negative outcome, reflecting recurrent illness. This directional dichotomy is reflected to varying degrees within different variables. When using a composite outcome for a prediction model, caution should be exercised to ensure that the model identifies individuals at risk of the intended outcomes of interest, rather than merely the proxies used to represent those outcomes. Identifying predictors with a consistent relationship for both outcomes may yield a more optimized and less biased prediction model for use in clinical care.

## Introduction

Neonatal mortality post-discharge is disproportionately high in low-and-middle-income countries (LMICs), particularly in sub-Saharan Africa and South Asia, and is caused largely by treatable infectious illnesses [1]. About one-third of deaths among newborns in LMICs occurs following hospital discharge. The high readmission rates and high mortality during subsequent readmission and at home after discharge reflects the high vulnerability of these neonates [2,3]. This vulnerable period offers an opportunity to use risk-guided models linked to proven interventions to optimize post-discharge care and improve outcomes [4].

Risk models frequently include mortality and readmissions as a composite outcome of interest in the development of algorithms and the evaluation of interventions [5–7]. In high-income settings, both readmission and mortality may be reasonably targeted together as indicators of adverse outcomes [8,9]. However, in low-and-middle-income settings, significant social and financial barriers to care seeking exist [10,11]. In fact, most pediatric post-discharge deaths do not occur during subsequent readmissions, but rather in the community. Thus, in LMICs, the ability to access care and hence be readmitted may not be dictated by illness severity alone, but also parental education of danger signs and socioeconomic disadvantages. In these settings mortality is high and readmission may have a greater impact on reducing mortality than HICs [12]. Therefore, composite prediction models using both post-discharge mortality and readmission as indicators of poor outcomes may yield misleading results.

Understanding the relationship between readmission and post-discharge mortality is therefore a prerequisite to not only building prediction outcome models, but also in guiding a more thorough understanding of the importance of readmission as both care-seeking behavior as well as an indicator of the severity of illness. This study aimed to investigate the effects of various maternal and newborn characteristics on mortality and readmission within the first six weeks following a facility delivery in Uganda.

## Methods

### Study design and setting

This prospective observational study was conducted between 14[th] July 2022 and 30[th] September 2023 which included a six-week post-discharge follow up period. Eligible mothers and newborns were enrolled after informed written consent from two regional referral facilities in Uganda: Mbarara Regional Referral Hospital in southwestern Uganda and Jinja Regional Referral Hospital in eastern Uganda. If the eligible participant was a minor, the parent or legal guardians(s) of the participant were approached first. During the consenting process, all women were free to withdraw from the study at any point without needing to give a reason for withdrawal.

This study was approved by Makerere University School of Public Health (MakSPH) Institutional Review Board (SPH-2021-177), the Uganda National Council of Science and Technology (UNCST) in Uganda (HS2174ES) and the University of British Columbia in Canada (H21-03709). This study has been registered at clinicaltrials.gov (NCT05730387).

### Study population

This was a secondary analysis of a study aimed at the development of predictive models to identify post-discharge outcomes for mothers and newborns following facility delivery to better guide post-natal care, and has been described in detail elsewhere [13]. Briefly, this study enrolled mothers and their newborn(s) who were admitted for delivery at two regional referral hospitals in Uganda. The Mbarara Regional Referral Hospital and the Jinja Regional Referral Hospital together serve a catchment area covering two districts and a population of approximately 973,000 individuals [13]. Together they provide a reasonable representation of the Ugandan population outside of the capital city of Kampala. These data were collected for the primary purpose of developing a post-discharge risk prediction model to identify high-risk dyads and subsequently promote interventions to reduce the risk of maternal and newborn morbidity and mortality.

### Data collection

Data collection tools are available through the Smart Discharges Dataverse [14]. Data were collected by trained study nurses during four periods: admission, post-delivery, before discharge, and six-week post-discharge follow-up. Data collected include pregnancy history, admission data, basic demographic details, delivery details of both the mother and newborn, and clinical data for mothers and their newborns discharged alive from the hospital. These variables were selected using a modified Delphi process based on clinical and contextual knowledge to identify candidate predictors for post-discharge mortality or readmission.

Six weeks following discharge, all mothers were contacted by phone for the follow-up data collection phase which captured outcomes including health seeking, re-admissions and deaths. Participants unreachable by phone were followed up in person by field officers. All data were collected using standardized questionnaires on encrypted study tablets and uploaded to a Research Electronic Data Capture (REDCap) database hosted at the BC Children's Hospital Research Institute (Vancouver, Canada) [15,16].

### Statistical methods

This analysis focused specifically on newborn outcomes. Thus, the primary outcomes for our analysis were six-week post-discharge mortality (all-cause) and six-week readmission in newborn infants, as well as a composite outcome of post-discharge mortality or readmission in the same period. Variables with missing data were imputed using multiple imputations through the Predictive Mean Matching (PMM) algorithm with 20 iterations as recommended for <1% power falloff [17]. Variables with only a single category recorded, and binary variables with low incidence (<50/6044 records) were omitted from the investigation.

Infant mortality and readmission were analyzed using survival analysis methods considering both outcomes as competing risks. The Cumulative Incidence Function (CIF) was estimated for both outcomes under the competing risk framework as the Kaplan-Meier (KM) estimate is biased in such cases [18,19]. Univariable Fine-Gray regression models for each outcome were then fit on each predictor, and the Subdistribution hazards Ratios (SHRs) were used to examine the effect of our predictors on outcome risk (associated with the instantaneous rate of occurrence in those subjects who are event-free or have experienced a competing event [20]). The composite outcome was analyzed using standard/classical survival analysis methods, with univariable Cox proportional hazards regression models fitted and Hazards Ratios (HRs) estimated for each predictor.

All analyses were conducted using R version 4.4.0 (R Foundation for Statistical Computing, Vienna, Austria) [21]. Competing risk analyses were performed using the 'cmprsk' package [22].

### Sample size and power

The sample size was determined based on criteria proposed by Riley *et al.*, 2020 [23] in view of the primary purpose of the dataset – building a prediction model for the composite outcome of death and readmission. Sample size estimates were calculated using the 'pmsampsize' R package [24]. The estimated sample size required was determined as 574 for the binary composite outcome.

As this was a secondary analysis, no formal sample size or power calculation was performed for the current exploratory analysis study. However, we note that the current study and potential future prediction models may not be sufficiently powered to analyze death and the rarity of the outcome was in fact the rationale for including readmissions as a composite outcome.

## Results

### Study population

A total of 7359 births were recorded, of whom 6044 newborns were discharged alive without admission to the neonatal intensive care unit and had complete follow-up (Fig 1). Thirty-five (0.58%) infants died within the six-week follow-up period, with median (interquartile range [IQR]) days to death of 10 (3–28). Two hundred and forty-one (3.99%) infants were readmitted within the same period, the median (interquartile range (IQR)) days to readmission being 15 (7–21) (Table 1).

The CIF for infant mortality and readmission as competing outcomes was calculated (Fig 2A). The probability of mortality (95% confidence interval (CI)) within the 6-week follow-up period was 0.005 (0.004–0.008). The probability of readmission (95% confidence interval (CI)) within the 6-week follow-up period was 0.040 (0.035–0.045). The Kaplan-Meier (KM) estimates for the combined outcome (95% confidence interval (CI)) within the 6-week follow-up period was 0.044 (0.036–0.051) for females, and 0.045 (0.038–0.053) for males (Fig 2).

### Infant mortality and readmission

Of the 206 predictor variables, 125 had opposite associations with the two outcomes, while 81 predictors had a consistent association with both outcomes (S1 Table). However, the degree of divergence in the effect measure varied widely across predictors.

Among the variables that had a consistent association with both outcomes, a diagnosis of HIV in the mother before/during pregnancy (Mortality SHR [95%CI]: 2.03 [0.71–5.78], Readmission SHR [95%CI]: 1.11 [0.68–1.82]), a medical history of high blood pressure in the mother (Mortality SHR [95%CI]: 3.07 [0.42–22.40], Readmission SHR [95%CI]: 1.66 [0.63–4.43]), and seeking antenatal care from a traditional birth attendant (Mortality SHR [95%CI]: 1.55 (0.21–11.34), Readmission SHR: 3.13 [1.84–5.34]) were associated with an increased risk of both mortality and readmission (Fig 3A).

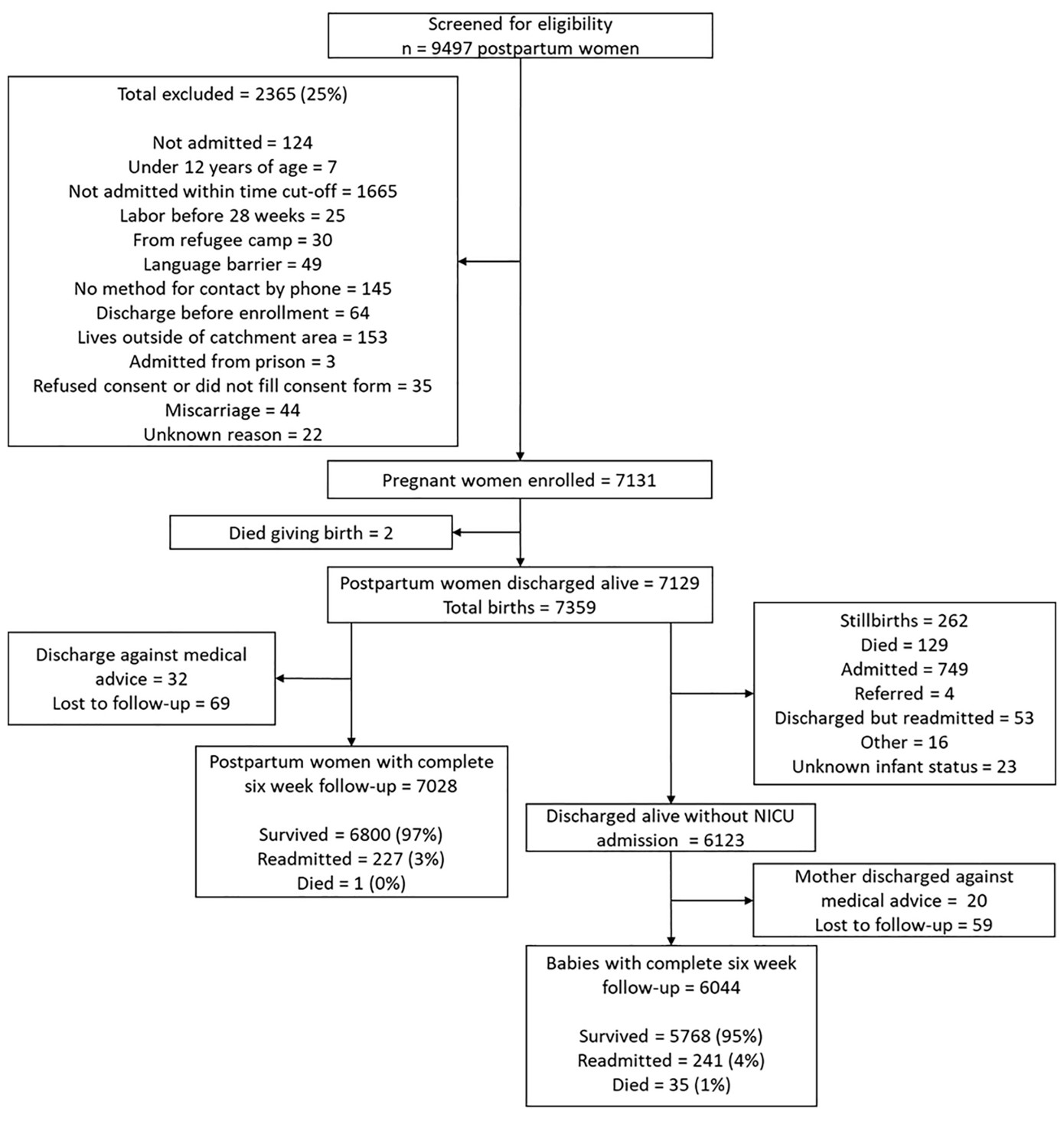

**Fig 1. Study enrollment flow diagram.**

**Table 1. Baseline Characteristics at admission for eligible mothers (N=7028) and newborns (N=6044).**

| Variable | N (%), Mean (SD), or Median (Q1, Q3) | N Missing (%) |
|---|---|---|
| Age of mother, years | 26.6 (6) | 0 (0%) |
| Highest level of education of the mother | | 3 (0.04%) |
| ≤P3 | 252 (3.6%) | |
| P4-P7 | 2228 (31.7%) | |
| S1-S6 | 3158 (44.9%) | |
| Post secondary (including post S4 technical school) | 1387 (19.7%) | |
| Household member undernourished during the pregnancy | 1218 (17.3%) | 3 (0.04%) |
| History of child death | 845 (12%) | 2 (0.03%) |
| Socioeconomic index | 3 (2, 4) | 2 (0.03%) |
| Time to hospital | | 1 (0.01%) |
| Less than 30 minutes | 3366 (47.9%) | |
| 30 minutes – 1 hour | 2672 (38%) | |
| More than 1 hour | 989 (14.1%) | |
| Mother Diagnosed with condition before pregnancy[a] | 851 (12.1%) | 0 (0%) |
| Mother Diagnosed with condition during pregnancy[b] | 4228 (60.2%) | 0 (0%) |
| HIV diagnosis before/during pregnancy | | 0 (0%) |
| No HIV | 6584 (93.7%) | |
| HIV Before Pregnancy | 379 (5.4%) | |
| HIV During Pregnancy | 65 (0.9%) | |
| Number of ANC visits attended | | 2 (0.03%) |
| <4 | 1830 (26%) | |
| 4-8 | 5073 (72.2%) | |
| >8 | 123 (1.8%) | |
| Caesarean delivery | 4496 (64%) | 1 (0.01%) |
| Gestation period <37 weeks | 586 (8.3%) | 0 (0%) |
| Sex of the baby (Male) | 3057 (50.6%) | 120 (1.99%) |
| Birth weight, kg | 3.2 (2.8, 3.5) | 121 (2%) |
| Length at birth, cm | 49.6 (48.5, 50.4) | 138 (2.28%) |
| Resuscitation at delivery | | 134 (2.22%) |
| Not resuscitated | 4183 (69.2%) | |
| Resuscitation with oxygen | 61 (1%) | |
| Resuscitation without oxygen | 1666 (27.6%) | |

[a] Includes HIV, high blood pressure, prior infertility, diabetes, kidney disease, sickle cell, hepatitis B/C, tuberculosis, and chronic mental illness. [b] Includes gestational diabetes, eclampsia, gestational hypertension, antepartum hemorrhage, PPROM, preterm labour, malaria, HIV, urinary tract infection, tuberculosis, anemia, COVID-19, hepatitis B/C, mental health illness and other infections otherwise not specified.

Meanwhile, an older age of the mother of 18–35 years (with reference to <18 years) (Mortality SHR [95%CI]: 0.78 [0.10–5.78], Readmission SHR [95%CI]: 0.62 [0.32–1.20]), a greater length of the baby at delivery (Mortality SHR [95%CI]: 0.85 [0.67–1.07], Readmission SHR [95%CI]: 0.91 [0.84–0.99]) and a weight of the baby greater than 2.5 kg (Mortality SHR [95%CI]: 0.14 [0.07–0.28], Readmission SHR: 0.68 (0.48–0.97)) were associated with a decreased risk of both mortality and readmission (Fig 3B).

Briefly, variables associated with an increase in infant mortality risk but a decrease in readmission risk included a travel time to the hospital greater than 1 hour (Mortality SHR [95%CI]: 1.40 [0.59–3.35], Readmission SHR [95%CI]: 0.28

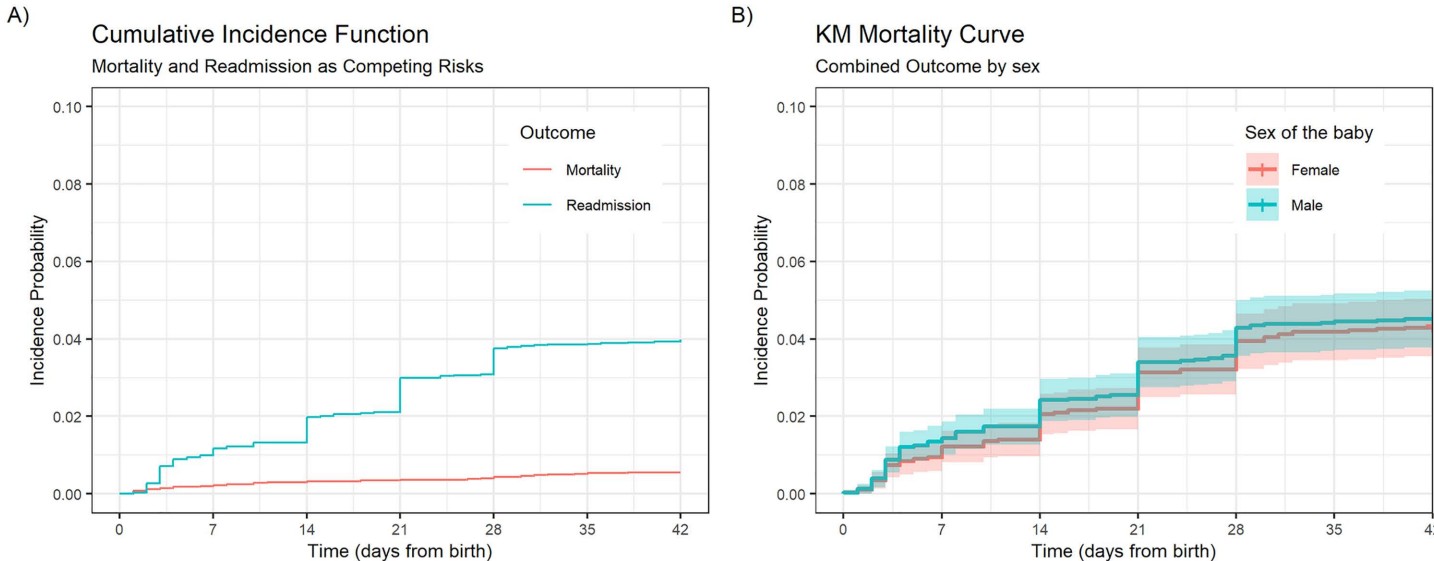

**Fig 2. Incidence of post-discharge mortality, readmission and combined outcomes. (A)** Cumulative Incidence of post-discharge mortality and readmission within the 6-week follow up period, estimated while treating both outcomes as competing risks. **(B)** Kaplan Meier (KM) mortality curve for the combined outcome of post-discharge mortality or readmission, grouped by sex.

[0.15–0.52]), caesarean-section mode of birth (Mortality SHR [95%CI]: 1.25 [0.62–2.50], Readmission SHR [95%CI]: 0.31 [0.22–0.44]), a diagnosis of ante-partum hemorrhage during the 3rd trimester (Mortality SHR [95%CI]: 3.22 [0.45–23.15], Readmission SHR [95%CI]: 0.43 [0.06–3.04]) and that of post-partum hemorrhage in the mother (Mortality SHR [95%CI]: 3.31 [1.01–10.85], Readmission SHR [95%CI]: 0.83 [0.37 to 1.86]), and twins/triplets being born (Mortality SHR [95%CI]: 2.12 [0.67–6.70], Readmission SHR [95%CI]: 0.46 [0.19–1.09]) (Fig 3C).

However, a higher household socio-economic index sum (Mortality SHR [95%CI]: 0.88 (0.73 to 1.06), Readmission SHR [95%CI]: 1.03 [0.96 to 1.10]), a history of previous child death (Mortality SHR [95%CI]: 0.57 [0.14 to 2.37], Readmission SHR [95%CI]: 1.21 [0.82 to 1.78]), resuscitation at birth (Mortality SHR [95%CI]: 0.14 [0.02 to 1.20], Readmission SHR [95%CI]: 2.71 [2.10 to 3.50]), a diagnosis of malaria (Mortality SHR [95%CI]: 0.44 [0.17 to 1.13], Readmission SHR [95%CI]: 2.12 [1.65 to 2.73]) or urinary tract infection (Mortality SHR [95%CI]: 0.22 [0.08 to 0.63], Readmission SHR [95%CI]: 1.82 [1.41 to 2.34]) of the mother during pregnancy are associated with a decrease in infant mortality risk, but an increase in readmission risk (Fig 3D).

The combined outcome of mortality and readmission behaves very similarly to readmission in terms of its association with different predictor variables, as seen in the HRs calculated using a Cox proportional hazards model (S1 Table). However, in those variables that have opposite directions of association with the two primary outcomes, the direction and magnitude of the point estimate of association often varied and was unreliable.

## Discussion

Our study shows that many predictors have conflicting directions and magnitudes of association while others reflect a consistent association in the same direction with the two outcomes when using a composite outcome of newborn post-discharge readmission and mortality. This is further evidenced by differences in pathways to readmission and post-discharge mortality as found in previous studies [25–27]. These associations are important for clinicians, researchers and policy makers to better understand the relationship between readmission and mortality in low-resource LMICs settings. Specifically, these data support the notion of readmission as both a beneficial outcome, reflecting appropriate

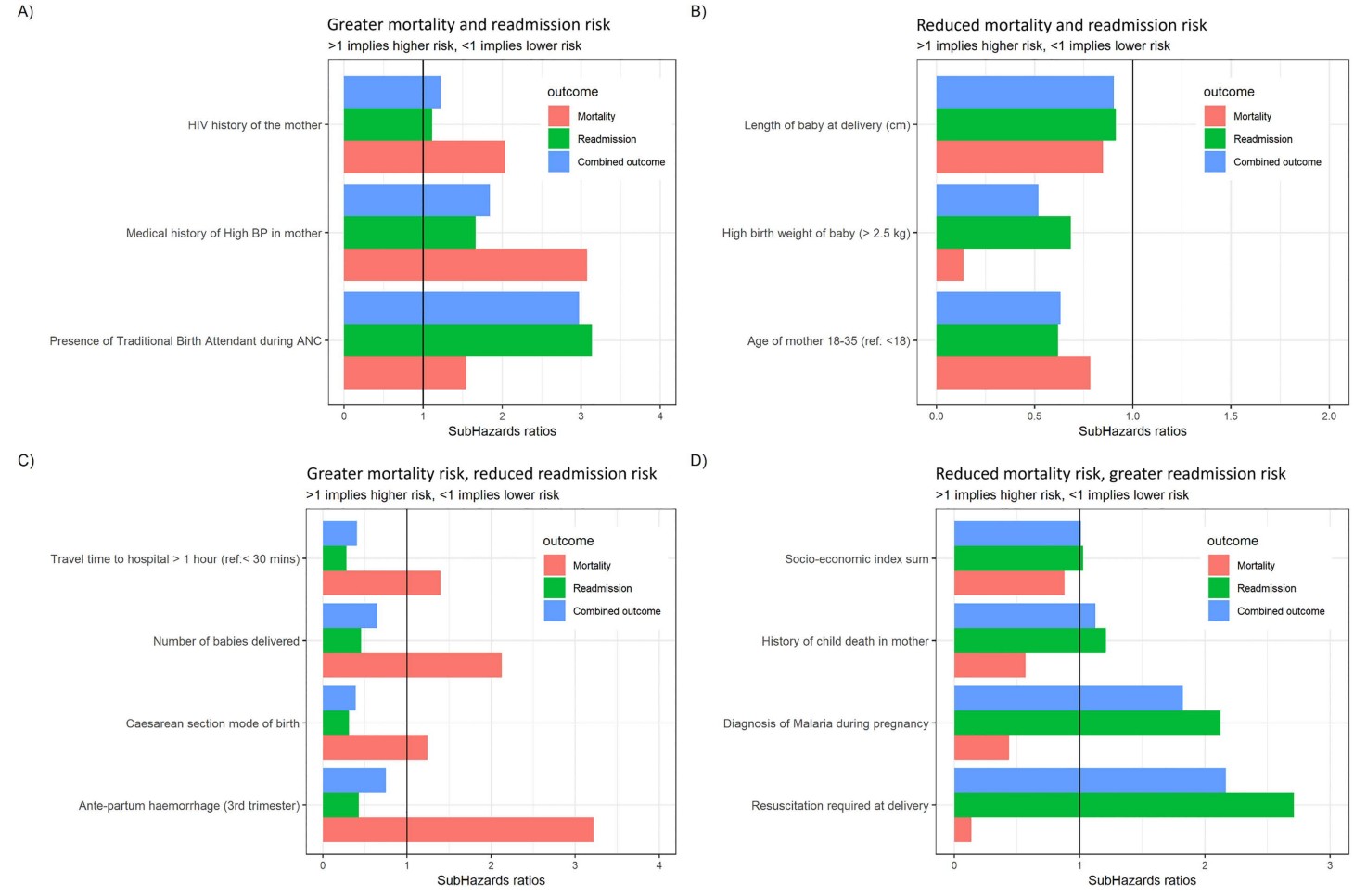

**Fig 3. Sub-hazards ratios for selected variables.** Variables that **(A)** increased both mortality and readmission risk. **(B)** decreased both mortality and readmission risk. **(C)** increased mortality but decreased readmission risk. **(D)** decreased mortality but increased readmission risk.

health-seeking behavior, as well as an adverse outcome, reflecting serious illness and possibly sub-optimal post-discharge care. The inconsistent associations bear considerable relevance in the use of readmission and mortality as a composite outcome in research and in surveillance metrics used to inform policy. If used together, an increased readmission rate coupled with decreased mortality may result in no or little change in the composite score and while beneficial to the patients, may be difficult to use as benchmark for improved outcomes.

Barriers to seeking care are well recognized as a key indicator of late presentation and poor health outcomes [28]. However, readmission is the culmination of a complex process and depends on recognition of the severity of illness, the resources to seek care, cultural preferences, power distance index between clinicians and patients, previous experience with the health system and the health system capacity to accommodate the patient. For example, we found that a travel time to hospital greater than 1 hour, and the baby being female both appear to increase mortality and decrease readmission suggesting barriers to care seeking. Higher socio-economic (SES) index as well as a discharge to the home of the mother appears to decrease mortality and increase readmission, suggesting fewer barriers to care. These observations are not unique to our study in that increased travel times or distance to healthcare facilities and sociocultural barriers linked to gender [29–32] hinder care seeking, while resources or support from other family members facilitate care

seeking and access [33–35]. Together with our results, these highlight the downstream effects that barriers to care and readmission can have on neonatal mortality.

Composite outcomes are often used in areas of medicine such as obstetrics, as mortality and severe morbidity may be very rare outcomes [36]. These have precedent [37], and advantages include addressing the multiplicity problem, increased statistical efficiency and easier interpretability [36,37]. However the combination of mortality and morbidity into a composite outcome is often inconsistently defined [38,39], and may not truly reflect mortality and other rare outcomes [40]. The use of outcomes like hospitalization which are more amenable to change also presents challenges in interpretation [40]. Similarly, readmission among neonates will pose challenges within a composite outcome. Further, such outcomes may not always adhere to basic statistical guidelines for composite outcomes, such as component outcomes being related directly to the primary objective, occurring with similar frequency and being of similar clinical severity [36,41–43]. Our data similarly suffers from mortality being infrequent, and the disagreement between the two outcomes.

As suggested by Dash et al. (2022) and Giles-Clark et al. (2023) [41,44], our study indicates that key component outcomes should be reported and assessed, though perhaps underpowered, for inconsistencies in predictive characteristics. Restricting the model to only those predictors that have a similar influence on both mortality and readmission may help ensure the composite outcome complies to the mentioned basic guidelines, and that any predictions stay clinically meaningful and interpretable [40,41,43].

Risk-differentiated care is increasingly called for to further improve child health outcomes in resource poor settings [45]. In the necessary data-driven approaches to define risk, models that stratify individual infants should be carefully defined, and efforts should be taken to include outcomes that are causally linked to illness or mortality, particularly if used as a composite risk measure. Well defined composite outcomes, however, if used appropriately, may be a powerful means of differentiating care according to the magnitude or drivers of risks, further improving healthcare system efficiency to improve child health in LMIC settings.

This study has several limitations. Our analysis was often not adequately powered for the rarer outcome of mortality, leading to wider confidence intervals and high p-values for many of the calculated hazard ratios. For multiple predictors, we have simply used the point estimate in the context of past results and a clinical understanding of the setting to infer associations. However, often it is this very sparsity in rare outcomes which necessitates the use of a composite outcome [36], and our data were originally collected with the composite outcome in mind. We use all-cause mortality and all-cause readmission for our analysis. Causes of death or readmission were not accounted for, though these may bear some relevance to the interpretation of our results. Within the short 6-week follow up period after birth, we anticipate most adverse outcomes in infants would be linked to pregnancy and infection related risk. Regardless, reasons for readmissions in particular are likely to be heterogenous in terms of cause and severity of illness and would provide useful insight regarding the care-seeking behavior of our cohort. Finally, our data did not capture motivations and barriers to health seeking which could further inform the analysis.

## Conclusion

Infant readmission within a LMIC setting is a complex outcome, reflecting not only illness but also care-seeking behavior. While there is a pressing need for approaches to adopting risk-differentiated care to aid in newborn post-natal care, a clear understanding of the motivators and barriers to readmission and their relation to mortality is important in developing such approaches.

## Supporting information

**S1 Table. Subhazards ratios for post-discharge neonatal mortality or readmission from Fine-Gray models, and hazards ratios for the composite outcome from Cox proportional hazards models.**
(DOCX)

**S2 File. PLOS One's inclusivity in global research questionnaire.**
(DOCX)

## Acknowledgments

We would like to thank individuals from the Smart Discharges Research Program (at Walimu, Kampala, Uganda and the Institute for Global Health at BC Women's and Children's Hospital, Vancouver, Canada) for their efforts in data collection, administration, logistics and study support including but not limited to: Clare Komugisha, Pascal M Lavoie, Marianne Vidler, Beth A. Payne, Jessica Trawin, Astrid Christofferson-Deb, Douglas Mwesigwa, Happy Annet Twinomujuni, Stefanie K. Novakowski, Dustin Dunsmuir, Abner Tagoola, Annet Mary Nabweteme, Bosco Busense, Miria Kantono, Nakasolo Moreen, Olivia Kamusiime, Jonan Nuwagaba, Kasande Clemensia, Turatemba Phionah, Fredson Tusingwire, Agaston Nuwarsasira, Christine Ankwatse, Livingstone Katsigaire, Haneen Amhaz, Obed Twinamatsiko, Peter Lewis, Charly Huxford, Bella Hwang.

## Author contributions

**Conceptualization:** Vuong Nguyen, Matthew O. Wiens.

**Data curation:** Yashodani Pillay, Angella Namala, Joseph Ngonzi, Nathan Kenya-Mugisha, Matthew O. Wiens.

**Formal analysis:** Abhroneel Ghosh.

**Funding acquisition:** J Mark Ansermino, Matthew O. Wiens.

**Investigation:** Abhroneel Ghosh, Vuong Nguyen, Matthew O. Wiens.

**Methodology:** Abhroneel Ghosh, Vuong Nguyen.

**Resources:** Vuong Nguyen, J Mark Ansermino.

**Software:** Abhroneel Ghosh, Vuong Nguyen.

**Supervision:** Vuong Nguyen, J Mark Ansermino, Matthew O. Wiens.

**Validation:** Abhroneel Ghosh.

**Visualization:** Abhroneel Ghosh.

**Writing – original draft:** Abhroneel Ghosh.

**Writing – review & editing:** Vuong Nguyen, J Mark Ansermino, Yashodani Pillay, Niranjan Kissoon, Matthew O. Wiens.

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
