## [Decision Letter · Decision Letter 0]

27 Oct 2025

Dear Dr. Ghosh,

Thank you for submitting your manuscript to PLOS ONE. After careful consideration, we feel that it has merit but does not fully meet PLOS ONE’s publication criteria as it currently stands. Therefore, we invite you to submit a revised version of the manuscript that addresses the points raised during the review process.

We look forward to receiving your revised manuscript.

Kind regards,

Chris A Rees, MD, MPH

Academic Editor

PLOS ONE

Journal Requirements:

“The Mitacs Globalink Research Internship provided the opportunity for Abhroneel Ghosh to conduct research with the team at the Institute for Global Health.”

4. Thank you for uploading your study's underlying data set. Unfortunately, the repository you have noted in your Data Availability statement does not qualify as an acceptable data repository according to PLOS's standards.

Additional Editor Comments:

-Minor point, but please change “increased” to “greater” on line 23 in the Abstract. Increase/decrease imply changes over time.

-Please include the total sample size and some basic demographics in the Results of the Abstract. Additionally, as the power to assess the relationship between post-discharge mortality and readmission is predicated upon the frequency of each event, please include the numbers for each outcome in the Abstract as well.

-Line 53: Access to healthcare is not “easy” for all patients and families in high-income settings. There are many documented disparities that make accessing care more challenging for certain populations. As such, please remove “easily”.

-As this study really relies on which candidate variables were collected to assess the relationship between readmission and post-discharge mortality, I request that the authors include more of a description of the rationale for the candidate variable selection in the Methods.

-Fig 3: please avoid increase/decrease as there is no relationship between the variables included and time in this figure. Greater and lower would be more appropriate.

Reviewer's Responses to Questions

**Comments to the Author**

1. Is the manuscript technically sound, and do the data support the conclusions?

Reviewer #1: Yes

Reviewer #2: Yes

Reviewer #3: Yes

2. Has the statistical analysis been performed appropriately and rigorously?

Reviewer #1: Yes

Reviewer #2: Yes

Reviewer #3: Yes

3. Have the authors made all data underlying the findings in their manuscript fully available?

Reviewer #1: Yes

Reviewer #2: No

Reviewer #3: Yes

4. Is the manuscript presented in an intelligible fashion and written in standard English?

Reviewer #1: Yes

Reviewer #2: Yes

Reviewer #3: Yes

Reviewer #1: This is helpful, well written manuscript that make a clear and decisive point. The question address is important, the methods used are appropriate and the interpretation is accurate to the results. I have read the manuscript in depth and do not believe the authors need to make any changes.

Reviewer #2: Check these out:

Rees CA, Kisenge R, Godfrey E, Ideh RC, Kamara J, Coleman-Nekar YJ, Samma A, Manji HK, Sudfeld CR, Westbrook A, Niescierenko M, Morris CR, Whitney CG, Breiman RF, Duggan CP, Manji KP. Derivation and Internal Validation of a Novel Risk Assessment Tool to Identify Infants and Young Children at Risk for Post-Discharge Mortality in Dar es Salaam, Tanzania and Monrovia, Liberia. J Pediatr. 2024 Oct;273:114147. doi: 10.1016/j.jpeds.2024.114147. Epub 2024 Jun 13. PMID: 38878962; PMCID: PMC11415288.

Rees CA, Kisenge R, Godfrey E, Ideh RC, Kamara J, Coleman-Nekar YJ, Samma A, Manji HK, Sudfeld CR, Westbrook A, Niescierenko M, Morris CR, Whitney CG, Breiman RF, Duggan CP, Manji KP. Derivation and Internal Validation of a Novel Risk Assessment Tool to Identify Infants and Young Children at Risk for Post-Discharge Mortality in Dar es Salaam, Tanzania and Monrovia, Liberia. J Pediatr. 2024 Oct;273

:114147. doi: 10.1016/j.jpeds.2024.114147. Epub 2024 Jun 13. PMID: 38878962; PMCID: PMC11415288.

Kisenge RR, Godfrey E, Ideh RC, Kamara J, Coleman-Nekar YJ, Samma A, Manji HK, Sudfeld CR, Westbrook A, Niescierenko M, Morris CR, Whitney CG, Breiman RF, Duggan CP, Manji KP, Rees CA. Development and Internal Validation of a Risk Assessment Tool to Identify Neonates at Risk for 60-Day Hospital Readmission in Dar es Salaam, Tanzania, and Monrovia, Liberia. Am J Trop Med Hyg. 2025 Apr 1;112(6):1378-1384. doi: 10.4269/ajtmh.24-0648. PMID: 40168977; PMCID: PMC12139531.

The discussion is shallow. there are many issues which have been breifly discussed, but not elaborated in detail. The travel time for example, may need more clarity. The reason for admission may also have various categories.

The maternal mortaality as a predictor of young infant mortaloty may also be considered.

Reviewer #3: This was an appropriately described analysis of composite outcome analysis in newborns in Uganda, recognizing the importance and potential utility of risk differentiation in clinical care in LMIC settings as well as the unique barriers to care. This manuscript was well written, easy to follow, with an appropriately described analysis. As a reviewer with limited knowledge of prediction modeling, I was able to follow along easily and appreciated the authors' clear writing and definitions. I also appreciated the authors recognizing the limitations of their data for interpretation and agree with the need to understanding the complex and multifaceted nature of barriers to readmission and mortality in the Ugandan (and other LMIC) context(s).

**Do you want your identity to be public for this peer review?** For information about this choice, including consent withdrawal, please see our Privacy Policy

Reviewer #1: No

Reviewer #2: No

Reviewer #3: No

---

## [Author Response · Author response to Decision Letter 1]

4 Jan 2026

We have enclosed our responses within the submitted "Response to Reviewers" rebuttal letter. We copy the same below for reference.

Editor

Minor point, but please change “increased” to “greater” on line 23 in the Abstract. Increase/decrease imply changes over time.

Response: Thank you for your careful review of our manuscript and for the insightful comments. We have appropriately changed increased/decreased to greater/reduced.

Please include the total sample size and some basic demographics in the Results of the Abstract. Additionally, as the power to assess the relationship between post-discharge mortality and readmission is predicated upon the frequency of each event, please include the numbers for each outcome in the Abstract as well.

Response: We have modified the Results of the Abstract to include the following lines:

Results: A total of 6040 newborns with complete six-week follow-up were enrolled, of whom 50.6% were male and 64% of mothers delivered via caesarean section. Thirty-three (0.55%) infants died within the six-week follow-up period and 241 (3.99%) were readmitted.

Line 53: Access to healthcare is not “easy” for all patients and families in high-income settings. There are many documented disparities that make accessing care more challenging for certain populations. As such, please remove “easily”.

Response: We have appropriately removed “where care is easily accessed.”

As this study really relies on which candidate variables were collected to assess the relationship between readmission and post-discharge mortality, I request that the authors include more of a description of the rationale for the candidate variable selection in the Methods.

Response: We have included the following line within our Methods section:

“These variables were selected using a modified Delphi process based on clinical and contextual knowledge to identify candidate predictors for post-discharge mortality or readmission.”

Fig 3: please avoid increase/decrease as there is no relationship between the variables included and time in this figure. Greater and lower would be more appropriate.

Response: Increase/decrease have appropriately been changed to Greater/lower.

Reviewer #1

This is helpful, well written manuscript that make a clear and decisive point. The question address is important, the methods used are appropriate and the interpretation is accurate to the results. I have read the manuscript in depth and do not believe the authors need to make any changes.

Response: Thank you for your careful review of our manuscript. We appreciate the recognition of the significance of our questions, and the confirmation of the relevance and rigor in our work. We are grateful for the positive review and that the manuscript meets expectations.

Reviewer #2

Check these out:

Rees CA, Kisenge R, Godfrey E, Ideh RC, Kamara J, Coleman-Nekar YJ, Samma A, Manji HK, Sudfeld CR, Westbrook A, Niescierenko M, Morris CR, Whitney CG, Breiman RF, Duggan CP, Manji KP. Derivation and Internal Validation of a Novel Risk Assessment Tool to Identify Infants and Young Children at Risk for Post-Discharge Mortality in Dar es Salaam, Tanzania and Monrovia, Liberia. J Pediatr. 2024 Oct;273:114147. doi: 10.1016/j.jpeds.2024.114147. Epub 2024 Jun 13. PMID: 38878962; PMCID: PMC11415288.

Rees CA, Kisenge R, Godfrey E, Ideh RC, Kamara J, Coleman-Nekar YJ, Samma A, Manji HK, Sudfeld CR, Westbrook A, Niescierenko M, Morris CR, Whitney CG, Breiman RF, Duggan CP, Manji KP. Derivation and Internal Validation of a Novel Risk Assessment Tool to Identify Infants and Young Children at Risk for Post-Discharge Mortality in Dar es Salaam, Tanzania and Monrovia, Liberia. J Pediatr. 2024 Oct;273

:114147. doi: 10.1016/j.jpeds.2024.114147. Epub 2024 Jun 13. PMID: 38878962; PMCID: PMC11415288.

Kisenge RR, Godfrey E, Ideh RC, Kamara J, Coleman-Nekar YJ, Samma A, Manji HK, Sudfeld CR, Westbrook A, Niescierenko M, Morris CR, Whitney CG, Breiman RF, Duggan CP, Manji KP, Rees CA. Development and Internal Validation of a Risk Assessment Tool to Identify Neonates at Risk for 60-Day Hospital Readmission in Dar es Salaam, Tanzania, and Monrovia, Liberia. Am J Trop Med Hyg. 2025 Apr 1;112(6):1378-1384. doi: 10.4269/ajtmh.24-0648. PMID: 40168977; PMCID: PMC12139531.

The discussion is shallow. there are many issues which have been breifly discussed, but not elaborated in detail. The travel time for example, may need more clarity. The reason for admission may also have various categories.

The maternal mortaality as a predictor of young infant mortaloty may also be considered.

Response: Thank you for your careful review of our manuscript. We are grateful for the provided links, as they provide useful information within the same context of post-discharge outcomes among infants in LMICs. We add these to our list of references, and cite them as follows:

“Our study shows that many predictors have conflicting directions and magnitudes of association while others reflect a consistent association in the same direction with the two outcomes when using a composite outcome of newborn post-discharge readmission and mortality. This is further evidenced by differences in pathways to readmission and post-discharge mortality as found in previous studies (25-27).”

For travel time, have expanded our discussion around this point to clarify its diverging role in reducing neonatal readmission whilst increasing mortality:

“These observations are not unique to our study in that increased travel times or distance to healthcare facilities and sociocultural barriers linked to gender (29–32) hinder care seeking, while resources or support from other family members facilitate care seeking and access (33–35). Together with our results, these highlight the downstream effects that barriers to care and readmission can have on neonatal mortality.”

We agree that the cause of readmission and mortality may play a role, however we were not able to include them in our analysis, and listed this as a limitation in the original text: “Causes of death or readmission were not accounted for, though these may bear some relevance to the interpretation of our results.” Further, we now also acknowledge that while 6-week readmission or mortality are likely to be pregnancy or infection related, reasons for readmissions are likely to be more heterogenous and have included the following text;

“Within the short 6-week follow up period after birth, we anticipate most adverse outcomes in infants would be linked to pregnancy and infection related risk. Regardless, reasons for readmissions in particular are likely to be heterogenous in terms of cause and severity of illness and would provide useful insight regarding the care-seeking behavior of our cohort.”

Finally, we agree that maternal mortality is likely a strong indicator of newborn mortality, however we only observed two in-hospital cases of mothers who died giving birth, and a single maternal death following discharge so it is not likely we would detect any meaningful associations. Hence, we did not make any edits with regards to this point.

Have the authors made all data underlying the findings in their manuscript fully available?

Reviewer #2: No

Response: We have appropriately created a new data repository on Borealis that adheres to the PLOS Data policy and PLOS One’s requirements: https://borealisdata.ca/dataset.xhtml?persistentId=doi:10.5683/SP3/QP5WYE

However, due to the sensitive nature of our data, and it being not fully de-identified, we choose not to allow unregulated access. Our updated Data Availability statement is as follows:

“Study materials (dataset, data dictionary, and metadata) are publicly available through the Pediatric Sepsis Data CoLaboratory’s (Sepsis CoLab) Dataverse: https://doi.org/10.5683/SP3/QP5WYE . Due to the sensitive nature of clinical data and the potential risk for re-identification of research participants, the de-identified dataset is available through moderated access. Access to this data will be granted on a case-by-case basis following approval from the authors and the Data Governance Committees.”

Reviewer #3

This was an appropriately described analysis of composite outcome analysis in newborns in Uganda, recognizing the importance and potential utility of risk differentiation in clinical care in LMIC settings as well as the unique barriers to care. This manuscript was well written, easy to follow, with an appropriately described analysis. As a reviewer with limited knowledge of prediction modeling, I was able to follow along easily and appreciated the authors' clear writing and definitions. I also appreciated the authors recognizing the limitations of their data for interpretation and agree with the need to understanding the complex and multifaceted nature of barriers to readmission and mortality in the Ugandan (and other LMIC) context(s).

Response: Thank you for your careful review of our manuscript. We are grateful that the relevance of our work on composite outcomes to clinical care in LMIC settings was recognized. It is also encouraging to learn that the results and writing are clear to an audience without detailed knowledge of statistical modeling, and that the text supports understanding across a broad interdisciplinary target audience including clinicians.

---

## [Editor Report · Decision Letter 1]

15 Jan 2026

Assessing the validity of post-discharge readmission and mortality as a composite outcome among newborns in Uganda

PONE-D-25-46967R1

Dear Dr. Ghosh,

We’re pleased to inform you that your manuscript has been judged scientifically suitable for publication and will be formally accepted for publication once it meets all outstanding technical requirements.

Kind regards,

Chris A Rees, MD, MPH

Academic Editor

PLOS One
---

## [Editor Report · Acceptance letter]

PONE-D-25-46967R1

PLOS One

Dear Dr. Ghosh,

I'm pleased to inform you that your manuscript has been deemed suitable for publication in PLOS One. Congratulations! Your manuscript is now being handed over to our production team.

Kind regards,

on behalf of

Dr Chris A Rees

Academic Editor

PLOS One